# Poisoning from *Alocasia × amazonica* Roots: A Case Report

**DOI:** 10.3390/toxins17040189

**Published:** 2025-04-10

**Authors:** Stanila Stoeva-Grigorova, Stela Dragomanova, Maya Radeva-Ilieva, Gabriela Kehayova, Simeonka Dimitrova, Simeon Marinov, Petko Marinov, Marieta Yovcheva, Diana Ivanova, Snezha Zlateva

**Affiliations:** 1Department of Pharmacology, Toxicology and Pharmacotherapy, Faculty of Pharmacy, Medical University of Varna, 84 “Tsar Osvoboditel” Blvd., 9000 Varna, Bulgaria; stela.dragomanova@mu-varna.bg (S.D.); maya.radeva@mu-varna.bg (M.R.-I.); gabriela.kehayova@mu-varna.bg (G.K.); simeonka.dimitrova@mu-varna.bg (S.D.); petko.marinov@mu-varna.bg (P.M.); snezha.zlateva@mu-varna.bg (S.Z.); 2Department of Urology, Faculty of Medicine, Medical University of Varna, 9000 Varna, Bulgaria; dr.marinov.simeon95@gmail.com; 3Clinical Toxicology Department, Naval Hospital, 9000 Varna, Bulgaria; marietyovcheva@gmail.com; 4Department of Biochemistry, Molecular Medicine and Nutrigenomics, Faculty of Pharmacy, Medical University of Varna, 9000 Varna, Bulgaria; divanova@mu-varna.bg

**Keywords:** *Alocasia × amazonica*, self-poisoning, intoxication, oxalates, poisoning management

## Abstract

All parts of *Alocasia* × *amazonica* (*A. amazonica*, Araceae) pose a toxicological risk due to oxalate production. Ingestion of the plant extract may cause multi-organ damage and fatal outcomes. Given the rarity of poisoning cases, its toxicological profile and systemic effects remain insufficiently characterized. This study aimed to investigate and report an appropriate approach to managing a patient intoxicated with *A. amazonica* (Araceae). A case of intentional self-poisoning with *A. amazonica* is presented. The patient, a 63-year-old woman, ingested approximately 200–300 mL of liquid prepared from the grated root of the plant. The initial clinical presentation involved localized injuries to the oral cavity and gastrointestinal tract, including severe pain, hoarseness, aphonia, dysphagia, mucosal erosions, and necrosis. Additional symptoms included hematinic vomiting, hemorrhagic diarrhea, and abdominal discomfort. These superficial and mucosal lesions resolved without the development of adhesions. Systemic effects comprised impaired consciousness indicative of encephalopathy, early metabolic acidosis, pulmonary edema with acute respiratory insufficiency, mild liver dysfunction, and hematuria. The therapeutic protocol for oral poisoning management was appropriate, leading to the patient’s discharge after 20 days of hospitalization.

## 1. Introduction

*Alocasia × amazonica* (*A. amazonica*), commonly referred to as the “African mask”, belongs to the Araceae family. The *Alocasia* genus is considered the largest within the family, comprising approximately one hundred species found across tropical, subtropical, and temperate regions worldwide, prized for their striking foliage [1,2]. However, the incidence of intoxications involving these plants is increasing, not only due to their ornamental value but also because certain species closely resemble their relatives, which are used for digestive or phytomedical purposes [3,4,5]. Similar to many other raphide-containing members of the *Alocasia* genus, *A. amazonica* produces oxalates as a primary metabolite. According to Diaz (2016), these metabolic products belong to the category of so-called gastrointestinal-hepatotoxic toxins [6]. Given the significant role of oxalate exposure in plant-induced toxicity, a specialized classification system categorizes plants into two distinct groups: Group A (containing insoluble oxalates) and Group B (containing soluble oxalates) (Table 1) [7].

Plants in Group A contain insoluble calcium and magnesium oxalates, which primarily lead to localized irritative–allergic reactions when the leaves are chewed or when sap comes into contact with the skin or eyes (Figure 1).

However, in animals that graze large quantities of leaf mass or stems, severe systemic poisoning has been observed. It is well established that ruminants exhibit greater resistance to oxalate toxicity (in comparison to monogastric animals) due to the presence of oxalate-degrading bacteria in their rumen, as well as the capacity for the formation of non-absorbable oxalate crystals, which are subsequently excreted directly in the feces. Nevertheless, excessive consumption of plants with high oxalate content may overwhelm the protective effect of microbial degradation, leading to hypocalcemia, nephrotoxicity, and acute renal failure. A low dietary calcium content may also contribute to renal failure [8]. Systemic effects in humans are similarly anticipated when a substantial mass of leaves, stems, roots, or their extracts are ingested. Group B comprises soluble oxalates, including oxalyl chloride, potassium oxalates, ammonium oxalates, and oxalic acid. These compounds do not induce local irritation, which consequently increases the risk of ingesting larger quantities, leading to systemic poisoning of varying degrees of severity [9,10,11].

The most well-known representatives of Group A, *Philodendron* and *Dieffenbachia* species, have been associated with more frequent poisoning cases. Krenzelok et al. (1996) and Pedaci et al. (1999) studied groups of patients who ingested parts of them and reported that only a small proportion of people experienced local damage from both plant species—15.5% and 18.2%, respectively [12,13]. No symptoms were reported among the remaining patients. In addition to localized irritation and gastrointestinal toxicity lasting up to 3 days, neurotoxicity has also been documented in cases of *Alocasia macrorrhiza* (giant elephant ear) poisoning, which was believed to be caused by the presence of sapotoxin in the tubers [14]. Another clinical study included 25 patients who ingested one or two bites of *Alocasia macrorrhiza* leaves and developed immediate symptoms. All cases were classified as mild or moderate. Half of the patients experienced numbness and paresthesia around and in the mouth, speech difficulties, and excessive salivation. Several patients also reported stomach aches and dysphagia [15].

Jadhav and Gugloth (2019) described a clinical case of a 4-year-old child who was poisoned after ingesting the roots of another Group A plant, *Arisaema triphyllum* [16]. Local toxic manifestations and respiratory complications were observed, attributed to the calcium oxalate crystals primarily present in the stem, leaves, and roots. Prakash et al. (2018) reported an incident with another plant from the Araceae family. It involved a 20-year-old man who ingested 50.0 g of *Arum maculatum* tuber with suicidal intent, leading to a diagnosis of angioedema [17]. On the other hand, fatal poisoning cases associated with the consumption of *Rumex crispus*, a plant containing soluble oxalates, have been reported. Toxic symptoms included vomiting, diarrhea, metabolic acidosis, severe liver and pulmonary damage with acute respiratory insufficiency, hypocalcemia, tetany, coma, and ventricular fibrillation, ultimately resulting in death. Autopsy findings revealed calcium oxalate precipitation in the brain and kidney vessels, as well as in the capillaries of the lungs, liver, and heart [9,18].

Additionally, it is well known that plants, including plant roots, are a source of various proteins, particularly proteolytic enzymes [19]. Members of the Araceae family such as *Alocasia* spp. also contain proteolytic enzymes [20]. Along with oxalates, these enzymes are shown to cause toxic effects in animals and humans and are released upon plant damage. Upon contact with tissues, proteolytic enzymes cause inflammation due to increased release of histamine and other proinflammatory mediators [21,22].

The diagnosis and management of poisonings induced by plants continue to present challenges for frontline clinicians, primarily due to the variability of clinical presentations and the infrequent nature of such incidents. In emergency settings, the lack of morphological identification and biochemical confirmation of plants also complicate immediate clinical intervention. Consequently, clinical features are crucial for initiating supportive treatment, which remains the primary approach for cases with uncertain etiology [23]. Thus, the present study aims to enhance healthcare professionals’ ability to recognize *A. amazonica* poisoning, focusing on its symptoms, diagnosis, and medical intervention. To address these gaps, we present our clinical experience in managing intentional self-poisoning with this plant.

## 2. Case Report

A 63-year-old female ingested approximately 200–300 mL of liquid prepared from grated *A. amazonica* root with suicidal intent. One hour later, she was admitted to the Clinical Toxicology Department at Naval Hospital, Varna, Bulgaria. Upon arrival, the patient was found in a distressed state, with vomit containing slimy substances nearby, significant foam around the mouth, and loss of control in the pelvic area. Her medical records indicate a history of liver cirrhosis of viral origin, present for the past four years, along with stable esophageal varices. Regarding her medical history, the patient was diagnosed with cervical cancer eleven years ago, which was successfully treated with radioactive cobalt. Clinical evaluations since 1999 have indicated that she is otherwise healthy.

### 2.1. Physical Examination

The patient was admitted to the clinic in a severely impaired general condition. She was initially responsive but quickly fell asleep, subsequently entering a soporific state. Her skin appeared pale, without evidence of caput medusae, prominent single vessel stars, or perioral cyanosis. Acrocyanosis and copious amounts of frothy, bloody material oozing from the mouth were observed. The respiratory rate was 20 breaths per minute, with numerous moist wheezes heard in both lungs. Cardiac function was within normal limits, with a measured heart rate of 96 beats per minute and blood pressure of 140/80 mmHg. Heart sounds were clear, without murmurs. There was no edema in the patient’s limbs. On palpation, the abdomen was diffusely painful. Incontinence of the pelvic region and bloody diarrhea were also noted.

A mesopharyngoscopy revealed necrotic damage to the uvula, soft palate (velum), and the bottom of the pharyngeal wall on the 36th hour. An indirect laryngoscopy identified ulcerative necrotic damage to the epiglottis crown, with intact vocal folds. A fibrogastroscopy was not performed. A chest X-ray showed interstitial edema, which progressed to alveolar edema after three days and persisted for six days. Ultrasound imaging confirmed the presence of small-nodular liver cirrhosis, with normal dimensions of the portal vein (*v. portae*) and splenic vein (*v. lienalis*) varices. The electrocardiogram indicated sinus rhythm, a horizontal position, and voltage signs of left ventricular overload.

### 2.2. Laboratory Findings

The patient’s laboratory examinations were conducted at the Clinical Laboratory of the Military Medical Academy-Varna. Blood gas and alkaline analysis revealed moderate metabolic acidosis. Upon admission, total bilirubin and urea levels were slightly elevated, while total protein content and serum calcium levels were decreased. Calcium oxalate crystals were detected in the urine. All other laboratory parameters were within normal limits. The results are presented in Table 2.

### 2.3. Clinical Course

#### 2.3.1. Respiratory Symptoms

The leading symptom at the time of the hospitalization, as well as during the first 6 days, was an acute respiratory insufficiency. During the first 24 h, symptoms of local irritative damage to the oral cavity and digestive tract, as well as severe acute lung failure, were observed. The patient suffered from dyspnea as well as central and peripheral cyanosis due to progressive pulmonary edema that was confirmed by auscultation and X-ray data. The bacteriological examination isolated *Citobacter diversus*, *Pseudomonas*, and *Klebsiella*. The pulmonary damage symptoms were controlled within 2 weeks.

#### 2.3.2. Gastrointestinal Symptoms

Local damage included severe gastrointestinal bleeding and ulcerative necrotic changes in the oral cavity. The patient vomited approximately 400–500 mL of clear blood twice, with a four-hour interval between episodes, after which the bleeding ceased. A single episode of bloody diarrhea occurred subsequently. Diffuse abdominal pain syndrome manifested around 48 h, which was relatively responsive to spasmolytics. By day 7, the ulcerative necrotic lesions had subsided, and by day 17, only a whitish plaque remained on the surface of the tongue. Nutritional intake was initiated on day 7 with liquid, mushy, and soft foods, gradually incorporating other food types into the diet. On day 20, X-ray contrast examination revealed no adhesions in the stomach or intestines, although esophageal varices in the cardiac region were detected.

#### 2.3.3. Other Symptoms

Aphonia, occurring despite intact vocal cords, manifested later (on the 8th day of intoxication) and persisted beyond the patient’s discharge, lasting until the 20th day post-poisoning. Hematuria was observed for 3 days, subsequently resolving spontaneously.

### 2.4. Treatment

Treatment included a gastric lavage with medicinal charcoal, placement of a Blakemore tube for 3 days, endotracheal intubation and mechanical ventilation for 24 h, and an anti-corrosion mixture for local treatment of the oral cavity (benzocaine, methylprednisolone, tetracycline, vitamin A, and nystatin). Pharmacological treatment comprised a capillary vessel protector—rutascorbine; a combination of antibiotics—amoxicillin/clavulanic acid, ceftriaxone, and amikacin; vitamin B; hepatoprotective drugs such as ademetionine, silymarin, and L-ornithine-L-aspartate; and glucose and electrolyte solutions. Calcium gluconate was used as an antidote to oxalic acid. Standard doses were used.

## 3. Discussion

The consumption of leaves from *A. amazonica, Dieffenbachia*, or other plants containing insoluble oxalates causes severe local pain due to calcium oxalate needles embedding in the tissues, leading to irritative injuries in the oral cavity [24,25]. Since the primary symptom is immediate pain, ingestion of large amounts of the plant is limited, thus reducing the risk of severe systemic toxicity. Consequently, it is generally believed that the damage is confined to the oral cavity. However, in the present case, due to the accompanying diseases (liver cirrhosis and esophageal varices), erosive activity was also observed on the esophagus, stomach, and intestines. The gastrointestinal injuries healed within 20 days without any strictures, which confirms the short-lasting corrosion action of the insoluble plant oxalates. Simultaneously, the patient’s conscious intent to inflict self-harm further contributed to the failure of the pain signal to deter ingestion of the toxic substance, thereby facilitating the manifestation of systemic toxicity.

Another important sign of intoxication with plants from the Araceae family is the local allergic reaction, which may be triggered by the release of proteolytic enzymes from the leaves or other plant parts. This leads to histamine release, resulting in edema and hyperemia [25,26]. When applied to the oral mucosa or skin, the juice may also cause histamine release; this was not registered in our case. The absence of an acute local allergic reaction may be explained with heating of the solution before ingestion. If so, the proteolytic enzymes were probably degraded. Instead, ulcerative necrotic damage to the pharyngeal and laryngeal tissues was observed 36 h after admission. These toxic reactions are more likely to be caused by the calcium oxalates contained in the plant [5]. Moreover, a specific local symptom appearing after ingesting toxic plants is perioral and oral paresthesia. This presents with numbness, loss of sensitivity, and dysphonia without obvious neurological damage [25]. Chan et al. (1995) described a case in which paresthesia disappeared in about 5 days. In the literature, a case is also reported in which a delay in symptoms was observed. On the eighth day of poisoning, when the patient’s condition was already improving and she began to speak, an aphonic voice appeared which persisted until discharge from the hospital. Laryngoscopy revealed intact vocal cords. Later, at home, the patient’s voice returned to normal. Clinically, aphonia is evaluated as a manifestation of a hysterical reaction, but the potential toxic effects of oxalates on nerve receptors must be taken into account [14]. Some authors believe that this neurotoxicity could be due to the effects of other toxic substances in the composition of the ingested plant [27]. Oral numbness and pain, as well as a need for endotracheal intubation due to airway obstruction, were reported in two cases of *Alocasia odora* ingestion [28].

Despite the classification by Bailey and Bailey (1976), which indicates that *A. amazonica* contains insoluble oxalates in its leaf mass and stems, it has been established that some *Alocasia* species also contain soluble oxalates in their sap [7,29,30]. Although systemic injuries due to intoxication with plants from the Araceae family are usually rare, when they do occur, they may lead to multi-organ damage [17,31,32]. For instance, the nervous system impairment can result in a comatose state. In severe cases of poisoning with soluble oxalate-containing plants, the development of cerebral edema with a fatal outcome was reported. Autopsy revealed severe vascular damage with calcium oxalate precipitation in the intima surrounded by mononuclear cells and neutrophils (symptoms of aseptic meningitis) [9,14,33,34]. In our case, somnolence, a slight quantitative disturbance of consciousness, was observed. In this context, investigating the contribution of the sapotoxin present in the plant to the observed clinical presentation is warranted. Tetany is considered a characteristic neurotoxic syndrome of this type of intoxication. Reig et al. (1990) demonstrated that 600.0 mg of oxalic acid binds total ionized calcium in the body. In the current case, no symptoms of hypocalcemia were manifested—both tetany and cardiac arrhythmias were absent [18].

Toxic liver damage induced by soluble oxalate-containing plants range from minor transient jaundice, described in a child who ingested 20.0–100.0 g of *Rhubab* leaves and stems, to severe centrilobular necrosis and death [9,35]. In our case, we concluded that, against the background of existing liver cirrhosis, the lesions were mild and without clinical manifestation, as laboratory data indicated mild cytolysis and transient hyperbilirubinemia. Scully et al. (1979) and Reig et al. (1990) reported different cases of poisoning following the ingestion of a fatal amount of *Rumex crispus*, another plant containing soluble oxalates. In both cases, the leading symptom was persistent interstitial and alveolar pulmonary edema, associated with the soluble oxalates present in the plant [18,33].

Since the kidneys are the target organ of the toxic effects of oxalates, urogenital damage such as hematuria, oxaluria, albuminuria, nephropathy, and acute renal failure are expected in these intoxications [9,13,36,37]. Oxalates are excreted unchanged in the urine [38]. In addition, a case of transient hematuria and oxaluria was described in a 16-year-old boy after ingestion of a certain quantity of red currants [39]. In the reported case, despite the applied antibacterial treatment, hematuria was observed on the eighth day of intoxication. This raises the possibility of an irritant toxic effect on the bladder mucosa during the elimination of calcium oxalates found in urine. In terms of differential diagnosis, it was considered a case of exacerbation of chronic hemorrhagic post-radiation cystitis for cancer treatment 11 years ago. The hematuria passed without sequelae.

Metabolic acidosis in acute intoxication with sorrel soup was also observed [9]. In the specific case, the established metabolic acidosis, with a pronounced decrease in the plasma bases one hour after ingestion, revealed the chemical–toxic origin of this syndrome. Thus, in conclusion, exposure to both insoluble and soluble oxalates should be considered not only in relation to the plant’s taxonomic classification but also in regard to the specific form in which the plant was consumed—whether as plant tissue, sap, maceration products, or other preparations.

Conservative treatment is the primary approach to *Alocasia* poisoning. To reduce oral pain, drinking 120–240 mL of ice water has been recommended [25]. After ingesting the plant, drinking milk may be helpful in order to precipitate soluble oxalate by combining it with calcium [37]. Precipitation of oxalates in the gastrointestinal tract may be facilitated by administering 1–2 g of calcium chloride or calcium gluconate, as well as several tablets of calcium carbonate, either orally or via a nasogastric tube [40]. Inducing vomiting and gastric lavage have not been recommended in cases with severe ulcerative damage of the gastrointestinal tract [25]. However, if the patient has ingested difficult-to-digest parts of a toxic plant, lavage can be performed [41]. Considering that the patient vomited slimy substances in the present case, the decision was made to conduct a gastric lavage. Although sodium bicarbonate is an established treatment for certain toxic ingestions, including salicylate overdose and sodium channel blocker toxicity, its efficacy in oxalate poisoning remains inadequately substantiated. Furthermore, its administration should be carefully managed due to the potential risk of promoting calcium oxalate crystal formation [42]. In case of an allergic reaction, antihistamines and steroids were provided in order to reduce the edema of the larynx [28]. However, the administration of an antihistamine is controversial for the management of poisoning related to the Araceae family because some studies failed to show a protective effect of antihistamine, and the involvement of histamine in edema remains unclear [38]. In addition, pretreatment with an antihistamine failed to reduce edema of the tongue in guinea pigs exposed to *Dieffenbachia*, a member of the Araceae family. However, some studies have shown a protective effect in animals [39]. Fochtman et al. (1969) found that a corticosteroid only delayed onset of the inflammation reaction in an animal study of *Dieffenbachia* poisoning [43]. Despite calcium oxalate being insoluble (in leaf consumption), large amounts of it are believed to dissolve in the stomach, forming oxalic acid. This acid can combine with calcium in the blood, potentially causing hypocalcemia, as well as renal and hepatic impairment. Therefore, calcium gluconate is considered a specific antidote [28].

After ingesting a considerable amount of *A. amazonica,* the patient should be monitored for hypocalcemia and related complications, such as muscle cramps, weak irregular pulse, hypotension, and cardiac arrhythmias. In this case, calcium chloride and calcium gluconate should be given along with fluids.

## 4. Conclusions

The present case underscores the intricate toxicological profile of *A. amazonica* ingestion, demonstrating both local and systemic effects. This case is particularly valuable due to the deliberate self-poisoning involving the plant’s roots, an occurrence rarely reported. While calcium oxalate crystals induce severe pain and localized irritation within the oral cavity, the presence of pre-existing gastrointestinal conditions in the patient admitted to our medical unit facilitated the development of more extensive erosive damage beyond the oral mucosa. The potential for systemic toxicity, particularly in vulnerable individuals with underlying health conditions, is further emphasized. Neurotoxic manifestations, including paresthesia and transient aphonia, suggest that the toxic effects of oxalates may extend beyond their well-documented local irritant properties. Additionally, the observed hematuria raises the possibility of a direct irritant effect of calcium oxalates on the urinary tract.

Although systemic complications following Alocasia ingestion are rare, the potential for severe outcomes—including renal, hepatic, and neurological involvement—necessitates a cautious approach. Conservative management remains the cornerstone of treatment, focusing on symptomatic relief and the maintenance of electrolyte balance, particularly the monitoring of complications related to hypocalcemia. Given the variability in toxicity among species of the Araceae family, further investigations are necessary to elucidate the full spectrum of toxic effects and to refine clinical management strategies.

The clinical case described demonstrates a rare intoxication with *A. amazonica* juice. It contains oxalic acid and calcium oxalate crystals causing local lesions, as well as systemic damage. Local symptoms included pain, hoarse voice, aphonia, dysphagia, edema and vesicles of the oral mucosa, ulcerations, necrosis, vomiting hematin matters, diarrhea with hemorrhagic character, and diffuse stomach pain. They resolved in about 20 days without adhesions. Established systemic injuries were manifested with consciousness disturbance (somnolence and toxic encephalopathy), early metabolic acidosis, pulmonary edema, as well as acute pulmonary insufficiency, minor liver damage, and hematuria.

Some of the recommendations formulated in treating cases of poisoning with plants containing calcium oxalate crystals include emergency protocols to prevent airway obstruction. In the current case described, the treatment applied followed the therapeutic protocol of the management of oral poisonings: a gastric lavage with medicine charcoal, infusions for increased oxalates excretion with urine, and antidote treatment with calcium gluconate to prevent hypocalcemia symptoms. We applied a symptomatic treatment to the upcoming local and systemic complications with antimicrobial, hepatoprotective, and capillaroprotective medicines.

## Figures and Tables

**Figure 1 toxins-17-00189-f001:**
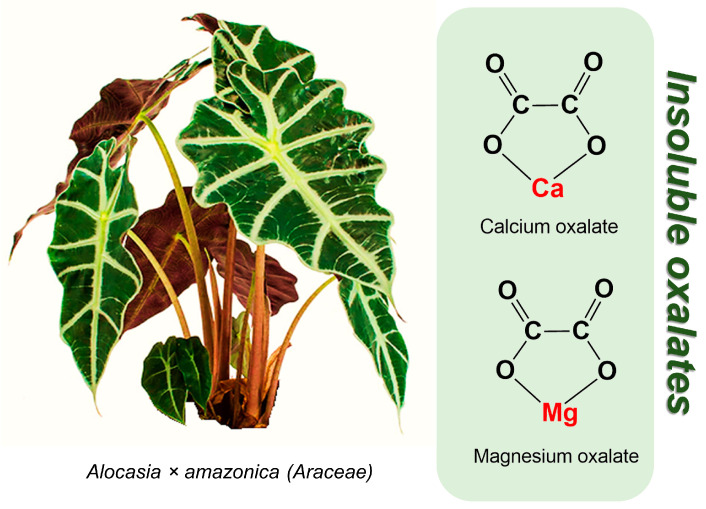
Chemical structures of calcium and magnesium oxalates produced by raphide *-containing species of the genus *Alocasia*. ***** raphide is a needle-shaped crystal of calcium oxalate.

**Table 1 toxins-17-00189-t001:** Plants containing oxalates.

Group A (Insoluble)	Group B (Soluble)
*A. amazonica*	*Parthenocissus tricuspidata (Boston ivy)*
*Agave* spp.	*Rumex crispus (garden sorrel)*
*A. macrorrhizos*	*Rheum (rhubarb)*
*Arisaema triphyllum (jack-in-the-pulpit)*	*Parthenocissus quinquefolia (Virginia creeper)*
*Achyranthes aspera*	
*Caladium* spp.	
*Colocasia* spp. *(elephant’s ear)*	
*Dieffenbachia* spp.	
*Halogeton glomeratus*	
*Monstera* spp.	
*Oxalis cernua*	
*Oxalis corniculata (creeping woodsorrel)*	
*Philodendron* spp.	
*Symplocarpus foetidus (skunk cabbage)*	
*Xanthosoma* spp.	
*Zantedeschia aethiopica (calla lily)*	

**Table 2 toxins-17-00189-t002:** Laboratory results.

	At Admission	Upon Discharge	Normal Range, Female
**Hemoglobin (g/L)**	139	109	121–151
**Hematocrit (%)**	44	27	36–48
**Leukocytes (g/L)**	12.4	8.6	4.5–11.0 × 10^9^/L
**Platelets (g/L)**	561	128	150–400 × 10^9^/L
**pH**	7.35	7.36	(arterial 7.35–7.45; venous 7.31–7.41)
**HCO_3_-act (mEq/L)**	18.8	24	22–28
**HCO_3_-std (mEq/L)**	19.4	24	22–28
**BE, base excess (mmol/L)**	6.8	1.1	−2–+2
**Aspartat aminotransferase (U/L)**	74	49	8–33
**Alanin aminotransferase (U/L)**	39	19	5–38
**Total bilirubin (µmol/L)**	56.7	38	1.71–20.5
**Amylase (U/L)**	160	87	30–110
**Total protein content (g/L)**	54	48.7	60–83
**Urea (mmol/L)**	12.1	8.5	1.8–7.1
**Serum calcium (mmol/L)**	1.2	2.4	2.2–2.7
**Calcium oxalate—urine**	+	*–*	

“+”—positive; “–”—negative.

## Data Availability

The original contributions presented in this study are included in the article. Further inquiries can be directed to the corresponding author.

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
