# Peer review of "Poisoning from Alocasia × amazonica Roots: A Case Report"

_toxins, 2025, doi:10.3390/toxins17040189_

Round 1
Reviewer 1 Report
Comments and Suggestions for Authors
Dear Editors,
This is a well written paper describing a single Alocasia poisoning combined with a review of human oxylate poisoning. It is well written and clearly presented. There is little reference to oxylate poisoning in animals which is still quite common. Though not essential to this discussion, a small suggestion might be to consider the species differences in oxylate poisoning and the differences in oxylate-induced diseases caused by different oxylate containing plants. As written this paper will be an excellent contribution to your journal.
Author Response
Comment 1:
This is a well-written paper describing a single Alocasia poisoning combined with a review of human oxylate poisoning. It is well written and clearly presented. There is little reference to oxylate poisoning in animals which is still quite common. Though not essential to this discussion, a small suggestion might be to consider the species differences in oxylate poisoning and the differences in oxylate-induced diseases caused by different oxylate containing plants. As written this paper will be an excellent contribution to your journal.
Response 1:
The species differences in oxalate poisoning have been discussed in the text.

Reviewer 2 Report
Comments and Suggestions for Authors
The article is very interesting and written at a very high level using appropriate references. The discussion is of very high quality.
Dear authors, I would like to give recommendation: "minor revision". However, I will not be able to see the correct version before publication. Therefore, I am forced to give the recommendation "mayor's revision". I am sorry.
I have several questions, comments and recommendations:
Please, provide the picture of plant and formula of oxalate.
- In line 37, you mentioned "ebidle relatives." Could you please provide some examples to clarify this term?
- Proteolytic enzymes are first mentioned in the discussion (line 181). In my opinion, they should also be introduced in the introduction. Do you believe that proteolytic enzymes were present in the poisoning liquid? If the liquid was prepared by cooking or using hot water, these enzymes would likely degrade. What is your perspective on this?
- In line 176, there is a statement about "weak corrosion action." I disagree because vomiting 500 ml of blood and experiencing bloody diarrhea suggest severe corrosion in my opinion. Perhaps "short-lasting corrosion action" would be more appropriate than "weak corrosion action“.
- The situation with calcium is intriguing. According to the table, hypocalcemia was observed at the beginning (1.4) and also after 20 days (1.2). However, in line 209, it is mentioned that there were "no signs of hypocalcemia." How is this possible? In line 239, drinking milk is recommended. What other treatments could be used, either orally or non-orally?
- In line 240, it is stated that lavage is not recommended. However, in line 159, it is confirmed that lavage was used. What is the correct procedure?
- Could you also critically discuss any incorrect or non-recommended methods? If doctors made mistakes, please highlight them and explain. This will help other doctors avoid making the same errors in the future. – this should be most useful part of this article.
Author Response
Comment 1:
In line 37, you mentioned "ebidle relatives." Could you please provide some examples to clarify this term?
Response 1:
Examples have been added to the text.
Comment 2:
Proteolytic enzymes are first mentioned in the discussion (line 181). In my opinion, they should also be introduced in the introduction. Do you believe that proteolytic enzymes were present in the poisoning liquid? If the liquid was prepared by cooking or using hot water, these enzymes would likely degrade. What is your perspective on this?
Response 2:
Proteolytic enzymes was mentioned in the Introduction section and the possibility for their content in the poisoning liquid was discussed in the Discussion section.
Comment 3:
In line 176, there is a statement about "weak corrosion action." I disagree because vomiting 500 ml of blood and experiencing bloody diarrhea suggest severe corrosion in my opinion. Perhaps "short-lasting corrosion action" would be more appropriate than "weak corrosion action“.
Response 3:
The suggestion has been accepted and implemented in the text.
Comment 4:
The situation with calcium is intriguing. According to the table, hypocalcemia was observed at the beginning (1.4) and also after 20 days (1.2). However, in line 209, it is mentioned that there were "no signs of hypocalcemia." How is this possible?
Response 4:
A typographical error was identified and subsequently corrected.
Comment 5:
In line 239, drinking milk is recommended. What other treatments could be used, either orally or non-orally?
Response 5:
Other treatments that could be used in oxalate poisoning have been discussed in the text.
Comment 6:
In line 240, it is stated that lavage is not recommended. However, in line 159, it is confirmed that lavage was used. What is the correct procedure?
Response 6:
In the Discussion section was discussed the reason for conducting a lavage.
Comment 7:
Could you also critically discuss any incorrect or non-recommended methods? If doctors made mistakes, please highlight them and explain. This will help other doctors avoid making the same errors in the future. – this should be most useful part of this article.
Response 7:
Some methods that are not recommended in oxalate poisoning have been discussed in the text.

Reviewer 3 Report
Comments and Suggestions for Authors
The report of poisoning by the roots of Alocasia x amazonica may represent an important technical-scientific contribution. However, the authors should readjust some parts of the manuscript.
Since the neurotoxic effects observed in this manuscript and described previously are associated with the consumption of roots, it would be interesting to specify in the title that this is root poisoning. Furthermore, in plants containing oxalate crystals, poisoning by roots is much less common than by leaves.
After the first mention of Alocasia x amazonica, the genus name should be abbreviated. The scientific name should always be written in italics. The name amazonica should appear in lowercase only.
A single case is not sufficient to indicate diagnostic and prognostic markers. Since many pathological factors, including the amikacin used in the treatment, cannot generate oxalate crystals in the urine, there is not enough strong evidence to associate Alocasia poisoning with the presence of oxalate crystals in the urine. Liver dysfunction is known to interfere with acid-base homeostasis, and liver cirrhosis often causes low calcium levels.
The conclusions section should be rewritten, as it mostly summarizes the case described, and the rest is speculative.
Specific comments:
L.67: include "leaves" after macrorrhiza
L.130-132: Figure 1 should be a table
L.163-164: What is the rationale for using antibiotics?
L.175-177: I did not understand why the plant was suspected of containing soluble oxalates. Injuries caused by insoluble crystals heal within a few days.
L.206-207: This effect was probably caused by sapotoxin.
L.213-216: Liver cirrhosis is not an acute lesion caused within hours.
L.250-253: Has oxalic acid ever been measured, or is this just speculation?
Author Response
Comment 1:
Since the neurotoxic effects observed in this manuscript and described previously are associated with the consumption of roots, it would be interesting to specify in the title that this is root poisoning. Furthermore, in plants containing oxalate crystals, poisoning by roots is much less common than by leaves.
Response 1:
The title has been revised.
Comment 2:
After the first mention of Alocasia x amazonica, the genus name should be abbreviated. The scientific name should always be written in italics. The name amazonica should appear in lowercase only.
Response 2:
The text has been revised in accordance with the provided comments.
Comment 3:
A single case is not sufficient to indicate diagnostic and prognostic markers. Since many pathological factors, including the amikacin used in the treatment, cannot generate oxalate crystals in the urine, there is not enough strong evidence to associate Alocasia poisoning with the presence of oxalate crystals in the urine. Liver dysfunction is known to interfere with acid-base homeostasis, and liver cirrhosis often causes low calcium levels.
Response 3:
The statement has been removed from the manuscript and the graphical abstract.
Comment 4:
The conclusions section should be rewritten, as it mostly summarizes the case described, and the rest is speculative.
Response 4:
The conclusions section has been rewritten.
Comment 5:
L.67: include "leaves" after macrorrhiza
Response 5:
The word “leaves” has been added to the text.
Comment 6:
L.130-132: Figure 1 should be a table
Response 6:
Table 2 has been generated and added to the text.
Comment 7:
L.163-164: What is the rationale for using antibiotics?
Response 7:
In the presentation of the case it is mentioned that a bacteriological examination was performed due to the respiratory symptoms and Citobacter diversus, Pseudomonas and Klebsiella were isolated. Thus, the patient was prescribed antibacterial medications.
Comment 8:
L.175-177: I did not understand why the plant was suspected of containing soluble oxalates. Injuries caused by insoluble crystals heal within a few days.
Response 8:
It should be “insoluble”. The mistake has been corrected.
Comment 9:
L.206-207: This effect was probably caused by sapotoxin.
Response 9:
The contribution of the sapotoxin present in the plant to the observed clinical presentation was mentioned in the text.
Comment 10:
L.213-216: Liver cirrhosis is not an acute lesion caused within hours.
Response 10:
We absolutely agree that liver cirrhosis is not an acute lesion. In the article is mentioned that the patient was diagnosed with liver cirrhosis of viral origin four years ago.
Comment 11:
L.250-253: Has oxalic acid ever been measured, or is this just speculation?
Response 11:
Oxalic acid was not investigated in the clinical case reported by us. The statement was made based on an analysis of the limited available literature. Nevertheless, the sentence was restructured to ensure it does not mislead the readers.

Round 2
Reviewer 2 Report
Comments and Suggestions for Authors
Authors did a great job, correct all necessary information and obtain new revelant information. However they do not provide picture of plant and formula of oxalates. Please add.
Author Response
Comment 1:
Authors did a great job, correct all necessary information and obtain new revelant information. However they do not provide picture of plant and formula of oxalates. Please add.
Response 1:
A visual representation of Alocasia × amazonica and the chemical formulas of oxalates has been included in the manuscript.
